# Exploring the Relevance between Gut Microbiota-Metabolites Profile and Chronic Kidney Disease with Distinct Pathogenic Factor

Tso-Hsiao Chen,[a,b,c] Chung-Yi Cheng,[a,b,c] Chun-Kai Huang,[d,e] Yi-Hsien Ho,[d,e] Jung-Chun Lin[e,f]

[a]Division of Nephrology, Department of Internal Medicine, Wan Fang Hospital, Taipei Medical University, Taipei, Taiwan
[b]Department of Internal Medicine, School of Medicine, College of Medicine, Taipei Medical University, Taipei, Taiwan
[c]Taipei Medical University-Research Center of Urology and Kidney (RCUK), School of Medicine, College of Medicine, Taipei Medical University, Taipei, Taiwan
[d]Department of Laboratory Medicine, Wan Fang Hospital, Taipei Medical University, Taipei, Taiwan
[e]School of Medical Laboratory Science and Biotechnology, College of Medical Science and Technology, Taipei Medical University, Taipei, Taiwan
[f]Pulmonary Research Center, Wan Fang Hospital, Taipei Medical University, Taipei, Taiwan

**ABSTRACT** The intimate correlation of chronic kidney disease (CKD) with structural alteration in gut microbiota or metabolite profile has been documented in a growing body of studies. Nevertheless, a paucity of demonstrated knowledge regarding the impact and underlying mechanism of gut microbiota or metabolite on occurrence or progression of CKD is unclarified thus far. In this study, a liquid chromatography coupled-mass spectrometry and long-read sequencing were applied to identify gut metabolites and microbiome with statistically-discriminative abundance in diabetic CKD patients ($n = 39$), hypertensive CKD patients ($n = 26$), or CKD patients without comorbidity ($n = 40$) compared to those of healthy participants ($n = 60$). The association between CKD-related species and metabolite was evaluated by using zero-inflated negative binomial (ZINB) regression. The predictive utility of identified operational taxonomic units (OTUs), metabolite, or species-metabolite association toward the diagnosis of incident chronic kidney disease with distinct pathogenic factor was assessed using the random forest regression model and the receiver operating characteristic (ROC) curve. The results of statistical analyses indicated alterations in the relative abundances of 26 OTUs and 41 metabolites that were specifically relevant to each CKD-patient group. The random forest regression model with only species, metabolites, or its association differentially distinguished the hypertensive, diabetic CKD patients, or enrolled CKD patients without comorbidity from the healthy participants.

**IMPORTANCE** Gut dysbiosis-altered metabolite association exhibits specific and convincing utility to differentiate CKD associated with distinct pathogenic factor. These results present the validity of pathogenesis-associated markers across healthy participants and high-risk population toward the early screening, prevention, diagnosis, or personalized treatment of CKD.

**KEYWORDS** chronic kidney disease, diabetes mellitus, fecal metabolite, gut microbiota, hypertension

Address correspondence to Jung-Chun Lin, lin2511@tmu.edu.tw.

The authors declare no conflict of interest.

The rising incidence and prevalence of chronic kidney disease (CKD) are considered essential health issues affecting around 10% of the global population (1). Hypertension, diabetes mellitus, or metabolic disorder is a widely-reported and critical risk factors toward the occurrence of CKD (2). Even though recent studies have documented the relevance of CKD progression with gender, age, or genetic signature in individual case (3, 4), persistent uncontrolled hyperglycemia, hypertension, or pro-fibrotic proteins is closely related to compromised kidney function or kidney injury, in turn resulting in end-stage renal disease

(ESRD) (5). It is an urgent need for new therapeutic strategy to halt or slow the progression of CKD and reduce the increasing morbidity and mortality in this population.

During past decades, the signature or impact of gut microbiota on diverse diseases, including CKD, has been continuously disclosed (6). Recent studies revealed the constant communication of gut microbiota with distinct organs of the host, including brain, immune system, kidney and nervous system, which maintained the physiological homeostasis (7–10). Gut microbial metabolism is one source of uremic toxins that lead to renal damage, CKD deterioration and occurrence of related complications (11–13). Gastrointestinal dysfunction has been widely noted in CKD patients, subsequently resulting in the reduced diversity or alteration in the microbial community that is different from healthy people (14). Decreases in the relative abundances of probiotics with the overgrowth of pathogenic bacteria, such as *Escherichia coli*, *Enterococcus*, or *Fusobacterium* genera, was characterized in the gut of CKD patients (15, 16). With proliferation of these pathogenic bacteria comes the production of uremic toxins, such as Trimethylamine N-Oxide and indoxyl sulfate, which in turn aggravating CKD and intestinal barrier injury (17, 18). In contrast, supplementation of *Lactobacillus* or *Bifidobacterium* genera was demonstrated to reverse gut dysbiosis and reprogram metabolite profile, in turn restoring the intestinal barrier integrity and reducing uremic toxins in CKD animal model (19, 20). Taken together, a comprehensive understanding of the gut microbiota-metabolite axis brings an emerging insight into the therapeutic strategy throughout the progression of CKD.

In this study, the structural change of gut microbiota and metabolite profile in the patients diagnosed with hypertensive CKD (H-CKD, $n = 26$), diabetic CKD (d-CKD, $n = 39$), CKD patients with no comorbidity condition (NC-CKD, $n = 40$) compared with healthy participants (HP, $n = 60$) was identified using Oxford Nanopore Technologies (ONT) long-read sequencer and LC-QTOFMS/MS platform. Launch of the sequencing platform developed by ONT achieves single molecule real-time sequencing toward microorganism genome with sequenced reads close to 2 Mb (21). The full-length *16S rRNA* gene sequenced with the innovation confers species-level resolution toward identification of bacteria (22). Identification of the pathogenic bacteria or metabolite composition exerts potential to serve an emerging marker for the early prediction, diagnosis, or prevention of CKD occurrence or deterioration. The impact of unique species or metabolite-mediated mechanism involved in pathogenesis of CKD is worthy of further pursue in the future work.

## RESULTS

**Characteristics of the enrolled CKD patients.** A total of 165 participants were enrolled in this study, and the cohort characteristics are summarized in Table 1. Compared with the enrolled H-CKD or NC-CKD patients, the d-CKD patients showed statistically-significant elevation in fasting glucose, HbA1c, and serum creatinine (Table 1, $P < 0.05$). Most H-CKD or NC-CKD patients were classified as stage 3 (Table 1, 67% and 73%), which was not noted among the d-CKD patients. There was no significant difference in the statistical results regarding estimated glomerular filtration rate (eGFR), age, or sex among all recruited CKD patients (Table 1).

**Statistical analysis of ONT sequencing results.** In this study, the genomic DNA extracted from the fecal sample of enrolled participants was subjected to the characterization of gut microbial communities by using the long-read sequencing platform (MinION, ONT, Oxford, UK). The CLC Genomics Workbench software (v.22.0.2; Aarhus, Denmark) was applied to evaluate the average number of sequenced reads and qualified reads per sample. No statistical difference regarding the sequencing efficiency was characterized among each group (Table 2, $P > 0.05$), whereas more OTUs were identified within the microbial communities in enrolled NC-CKD or d-CKD patients compared to those of H-CKD patients or healthy participants (Table 2, $P < 0.01$).

The analytic results of Simpson index (Fig. 1, left) or Shannon entropy (Fig. 1, right) indicated difference in the species diversity ($\alpha$-diversity) between the gut microbial communities in healthy participants and NC-CKD patients ($P < 0.005$) or d-CKD patients ($P < 0.005$), but not H-CKD patients ($P < 0.05$, Shannon entropy; $P > 0.05$, Simpson

**TABLE 1** Characteristics of healthy participants and enrolled diabetic, hypertensive CKD patients, or CKD patients without comorbidity

| Group | Healthy group (n = 60) | NC-CKD (n = 40) | Hypertensive CKD (n = 26) | Diabetic CKD (n = 39) | P |
|---|---|---|---|---|---|
| CKD stage No. (%) | –[a] | Stage 1 & 2: 5 (12.5%) Stage 3: 27 (67.5%) Stage 4 & 5: 8 (20%) | Stage 1 & 2: 4 (15.38%) Stage 3: 19 (73.07%) Stage 4 & 5: 13 (11.54%) | Stage 1 & 2: 10 (25.64%) Stage 3: 17 (43.6%) Stage 4 & 5: 12 (30.76%) | |
| Age (Median(IQR)) | 66 (41–87) | 69 (33–90) | 71 (40–88) | 71 (43–90) | |
| Sex (n,%) | | | | | |
| Female | 32 (53.33%) | 21 (52.5%) | 15 (57.69%) | 15 (38.46%) | >0.05 |
| Male | 28 (46.67%) | 19 (47.5%) | 11 (42.31%) | 24 (61.54%) | >0.05 |
| Fasting Blood Glucose (mg/dL) (Median(IQR)) | 89 (61–100) | 98 (77–157) | 99 (85–166) | 135 (84–425) ($P < 0.005$) | >0.05 |
| HbA1c (%) (Median(IQR)) | 5.1 (4.2–6.0) | 5.7 (4.1–6.4) | 5.9 (5.1–9.4) | 7.05 (5.6–9.4) ($P < 0.005$) | >0.05 |
| Serum Creatinine (mg/dL) (Median(IQR)) | 0.72 (0.5–1.15) | 1.4 (1.11–13.15) | 1.505 (0.97–3.85) | 1.615 (0.73–8.75) ($P < 0.05$) | >0.05 |
| eGFR (mL/min/1.73m2) (Median(IQR)) | 92.4 (63.9–134.2) | 40 (4–83) | 46 (16–69) | 40 (7–84) | >0.05 |

[a] –, not applicable.

index). Principal coordinates analysis (PCoA) with Weighted Unifract distance (Fig. 2, left) or Bray-Curtis indices (Fig. 2, right) was applied to visualize the dissimilarity of gut microbial communities among all groups. A Weighted Unifrac or Bray-Curtis distance between each sample was applied to indicate the discrimination in the gut microbial communities among healthy participants (Fig. 2, green dot) and H-CKD patients (Fig. 2, brown dot), d-CKD patients (Fig. 2, blue dot), or NC-CKD patients (Fig. 2, pink dot). The isolated clusters within microbial communities of each CKD-patient group were shown in PCoA space (Fig. 2). The beta diversity of the gut microbial communities differed between each group of the CKD patients compared with another or healthy participants (Fig. 2, PERMANOVA P value = 0.023 or 0.017).

**Classification of gut dysbiosis of the enrolled CKD patients.** A total of over 1,000 OTUs at the species level were classified with the usage of ONT sequencer and CLC Genomics Workbench software in this study. The top 25-ranked OTUs were shown based on the average reads that classified among all recruited group (Fig. 3, bar chart). Linear discriminant analysis (LDA) effect-size (LEfSe) analyses were conducted to discriminate the differential abundances of identified OTUs between healthy participants and CKD patients (23). The results of LDA score indicated statistically high levels of *Escherichia marmotae*, *Fusobacterium mortiferum*, *Streptococcus pasteurianus*, *Bacteroides stercoris*, *Lactobacillus mucosae*, *Culturomica massiliensis*, and *Subdoligranulum variabile* in the microbial communities of CKD patients (Fig. 3, right, red bar) compared with the healthy group (LDA score (log 10) < -3). In contrast, *Mitsuokella jalaludinii*, *Megasphaera indica*, *Selenomonas ruminantium*, and *Anaerostipes hadrus* were relatively more abundant in the gut microbiota of healthy group (Fig. 3, right, green bar; LDA score (log 10) > 3) compared to those of enrolled CKD patients. Among the CKD-related candidates, the higher abundances of *Streptococcus pasteurianus*, *Bacteroides stercoris*, and *Culturomica massiliensis* were specifically identified in the gut microbial communities of NC-CKD patients than those of other CKD patients (Fig. 4, left, red character), whereas the relatively high

**TABLE 2** Statistical summary of sequencing throughput and identified result in each enrolled group

| Group | Healthy group (n = 60) | NC-CKD (n = 40) | Hypertensive CKD (n = 26) | Diabetic CKD (n = 39) | P |
|---|---|---|---|---|---|
| No. of Raw reads per sample | 38,786 (±2,079) | 49,532 (±3,709) | 48,603 (±4,719) | 43,997 (±4,217) | >0.05 |
| No. of qualified reads per sample | 32,527 (±2,226) | 45,289 (±3,114) | 45,321 (±3,134) | 40,328 (±3,050) | >0.05 |
| Reads in identified taxa | 25,329 (±1,845) | 35,677 (±2,851) | 37,951 (±2,977) | 32,375 (±2,505) | >0.05 |
| Correctly classified (% (SD)) | 77.87 (±3.64) | 78.78 (±6.07) | 83.74 (±6.52) | 80.28 (±5.69) | >0.05 |
| No. of identified taxa per sample | 1,219 | 1,965($P < 0.01$) | 1,021 ($P > 0.05$) | 1,865 ($P < 0.01$) | |

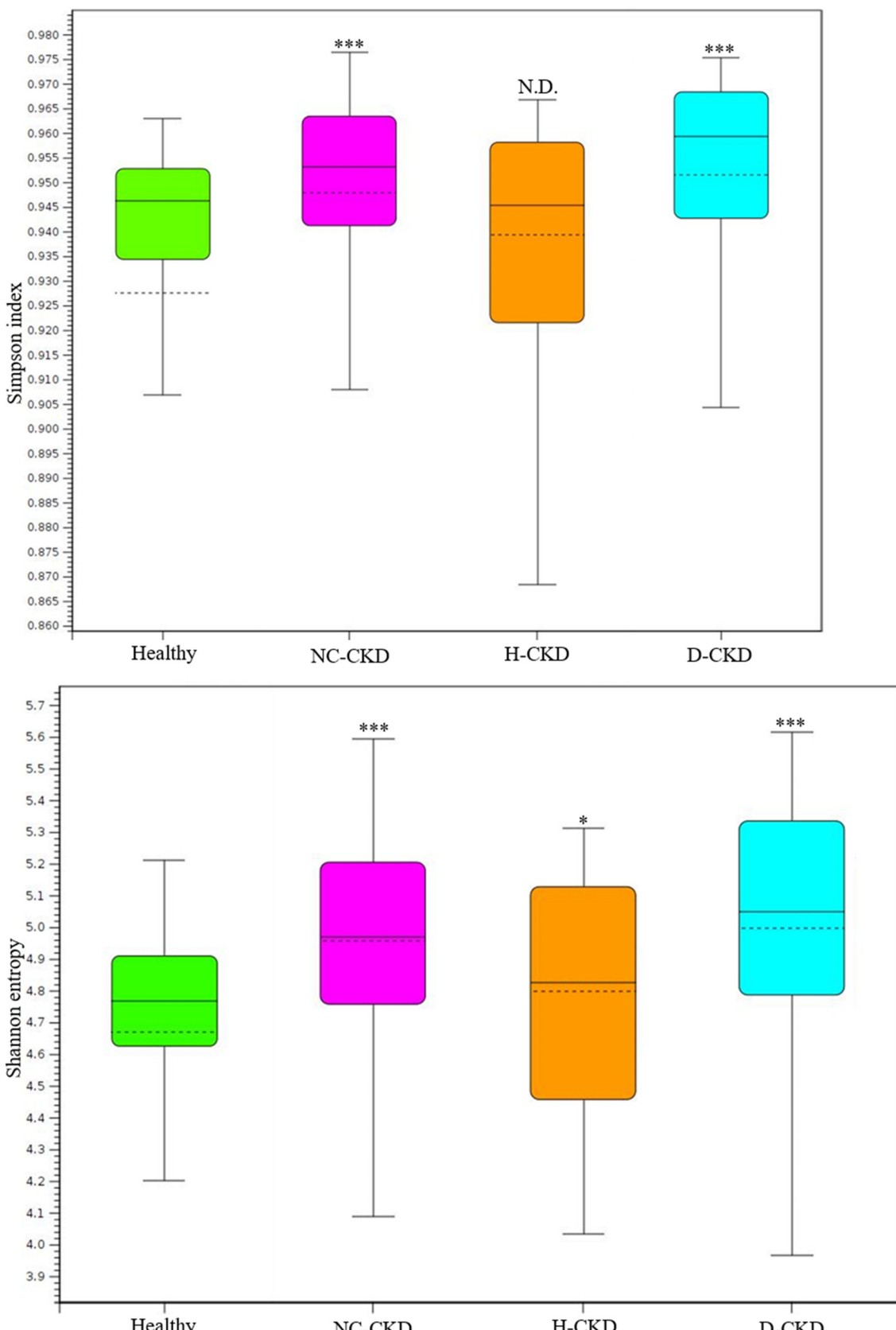

**FIG 1** Complexity of taxonomic composition among the enrolled groups with long-read sequencing results. The $\alpha$-diversity in all groups is synchronously evaluated using Simpson index (top) and Shannon entropy (bottom).

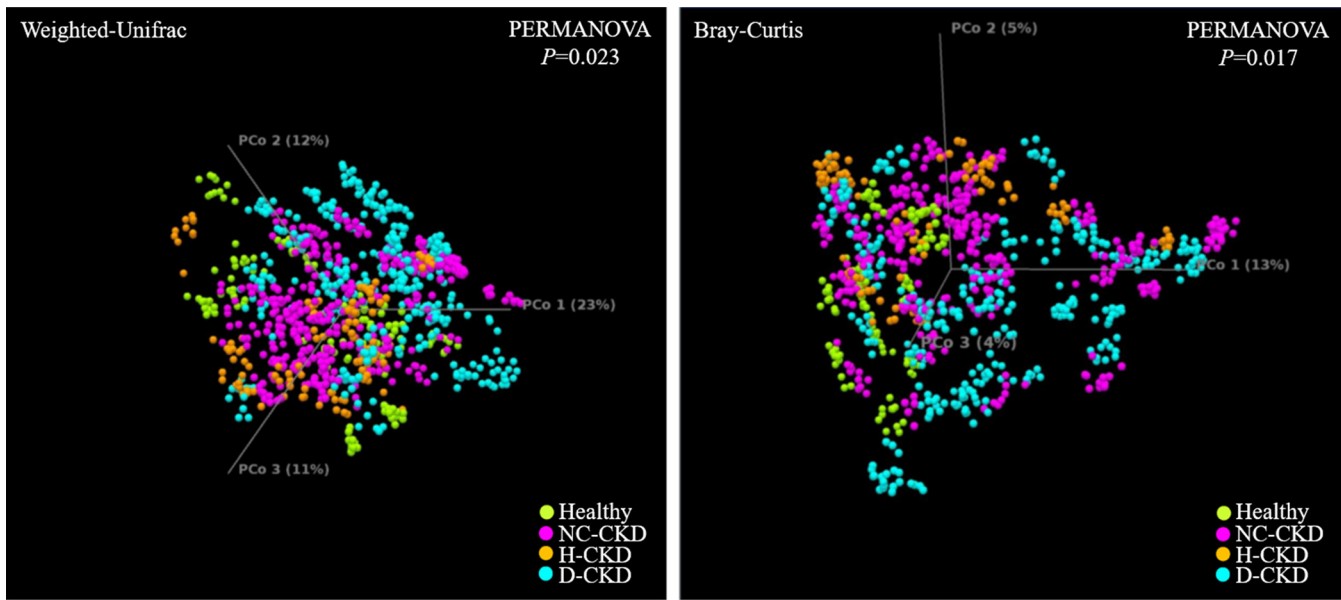

**FIG 2** Dissimilarity in taxonomic composition among the enrolled groups is evaluated by using Weighted Unifrac principal-component analysis (PCoA) (left) and Bray-Curtis dissimilarity analysis (right).

abundance of *Escherichia marmotae* was identified throughout the all CKD groups. The relatively abundant *Fusobacterium mortiferum* and *Lactobacillus mucosae* were specifically identified in the gut microbial communities of d-CKD patients (Fig. 4, right, red character). These results suggested the relevance between the structural changes of gut microbiota and the pathogenesis of CKD with discriminative disease condition.

**Discriminative metabolomic signatures of each CKD-patient group.** In this study, a total of 187 gut metabolites were identified in the fecal samples of enrolled participants by using the UPLC-MS/MS platform and corresponding analytic pipeline. The principal-component analysis (PCA) was conducted to evaluate the divergence of gut metabolite profiles among all groups. As shown in Fig. 5, the isolated clusters within metabolite profiles of each CKD group were identified in PCA space (Fig. 5, PERMANOVA, $P = 0.001$). The gut metabolites with discriminative abundances among healthy participants and each CKD group were characterized with the following criteria, including a significant alteration

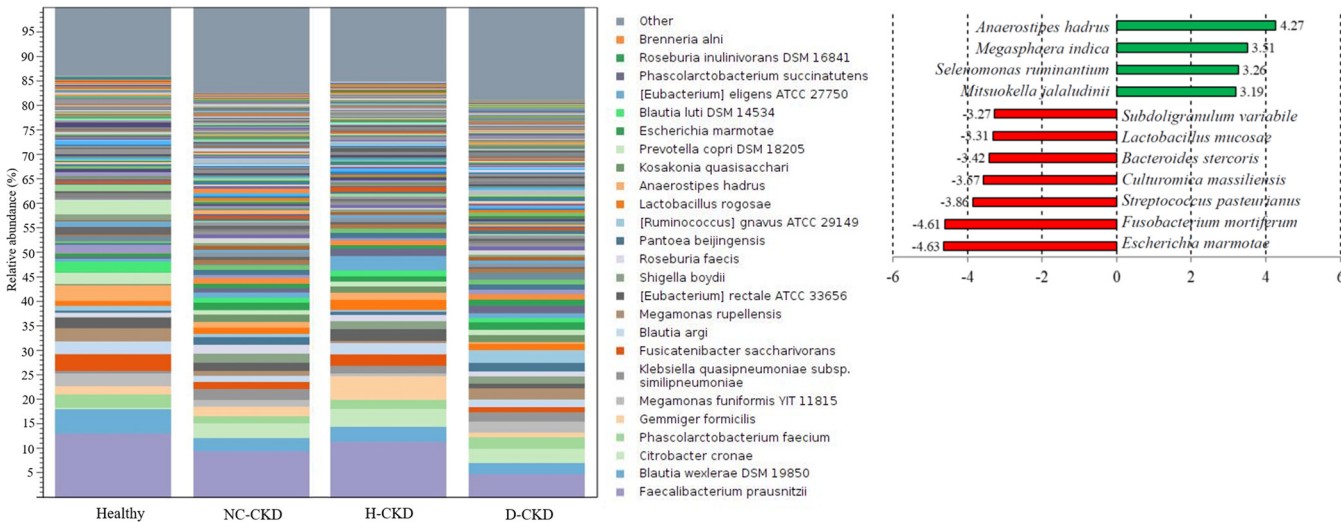

**FIG 3** Classification of operational taxonomy unit (OTU) with long-read sequencing in healthy participants and enrolled CKD patients. The relative levels of top 25 OTUs to species level based on the average reads among all recruited group is presented in stacked bar chart (left). Linear discriminant analysis (LDA) scores indicate discriminative abundances of OTUs in healthy participants (green bar) and CKD patients (red bar) (right).

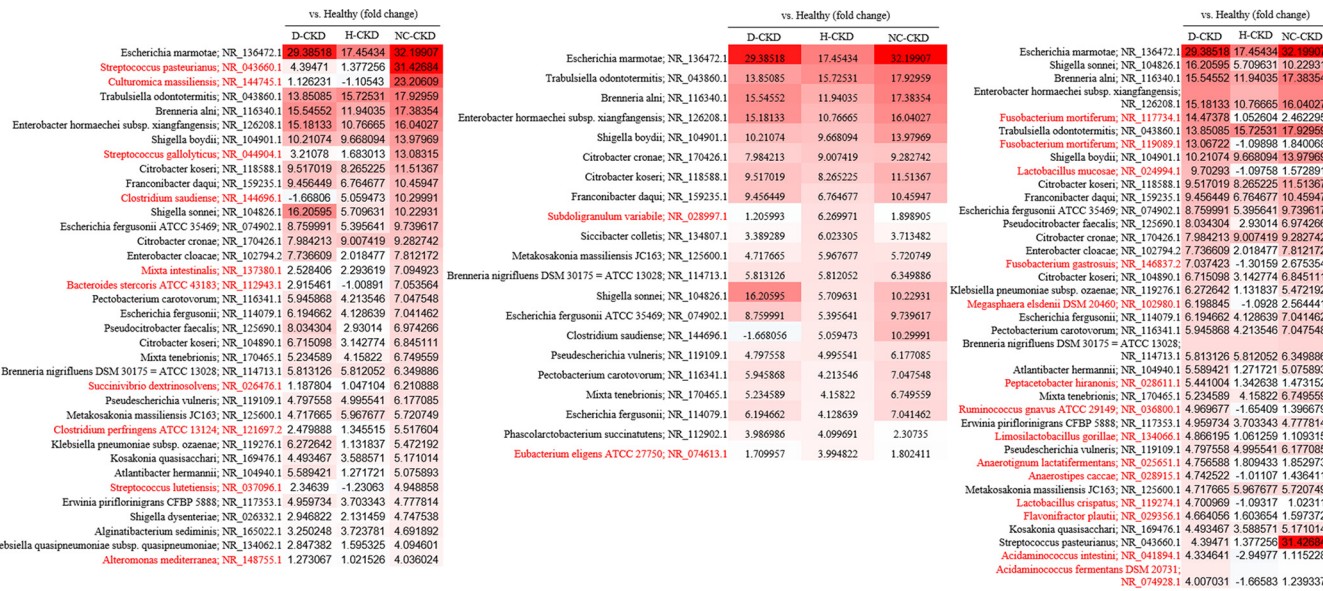

**FIG 4** Comorbidity-associated gut microbial communities across all enrolled CKD groups. Relevance between the discriminative abundance of classified OTUs at the species level and recruited CKD patients with distinct comorbidity is shown in a heat map chart (red character).

in relative abundance (-2> fold change >2), a variable importance in projection value (VIP) > 1.5, and a statistical $P$ value < 0.05 (Table 3). A heat map was shown to illustrate the increased abundances of 29 metabolites (Fig. 6, upper, red character) and decreased levels of 32 metabolites in each CKD group (Fig. 6, lower, blue character). These results suggested the potential application of identified metabolite for serving the specific marker toward the occurrence of CKD with distinct pathogenic factor.

To evaluate the impact of altered gut microbiota or metabolite on pathogenesis of CKD, the association between altered metabolite profile and microbial community identified in each CKD group was demonstrated with the utilization of the Zero-inflated negative binomial (ZINB) regression (R package pscl) (24). Among the identified OTUs or metabolite discriminating in distinct CKD group from healthy participants, the significant associations between *Streptococcus*, *Clostridium*, *Culturomica*, and *Bacteroides* genera and 4 NC-CKD-enriched metabolites, including Arachidic acid, L-Phenylalanine, Dihomo-gamma-linolenic acid, and N-Acetylputrescine were identified with NC-CKD occurrence (Fig. 7, left, $P$ < 0.05). The convincing correlations of *Fusobacterium* genera, *Megasphaera elsdenii*, *Ruminococcus gnavus*, and *Lactobacillus* genera with L-Proline and Stearic acid were identified to discriminate the d-CKD patients from the healthy participants (Fig. 7, right, $P$ < 0.05). In H-CKD groups, the close associations of relatively abundant Stearic acid, Amiloride, and 3,4-Dimethoxyphenylethylamine with the identified OTUs, including *Escherichia marmotae*, *Enterobacter hormaechei*, *Shigella boydii*, *Citrobacter koseri*, and *Subdoligranulum variabile* were identified (Fig. 7, middle, $P$ < 0.05). These results suggested the potential relevance between species-OTU associations and the diagnosis of CKD with distinct pathogenic factor.

**Predictive utility of species-metabolite association to the causation between CKD and risk factor.** The utility of characterized species-metabolite association on distinguishing CKD patients ($n$ = 96) from healthy counterparts ($n$ = 60) that recruited in our previous study was subsequently evaluated by using a random forest regression model (25). The results of receiver operating characteristics (ROC) curve were generated with the relative abundance of identified OTUs, CKD-enriched metabolites, or the strength of OTU-metabolite association in individual CKD group. The relevant OTUs discriminated NC-CKD ($n$ = 40), H-CKD ($n$ = 26), or d-CKD patients ($n$ = 30) from healthy group with an area under the ROC curve (AUC) from 0.645, 0.467, or 0.595 (Fig. 8, left). The utility of CKD-enriched metabolites at distinguishing distinct CKD groups from healthy participants were identified with an AUC from 0.714, 0.841, or 0.701 (Fig. 8,

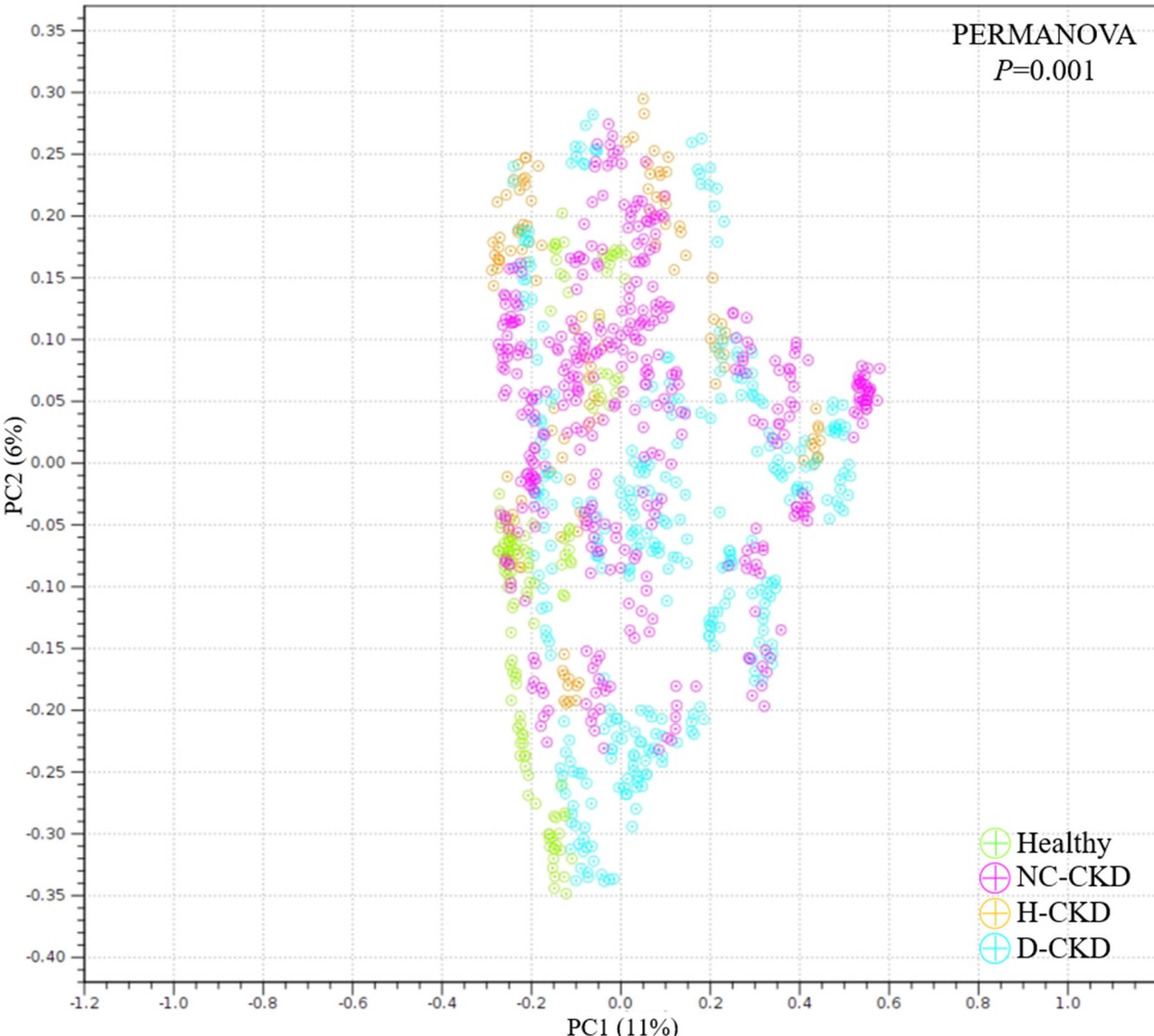

**FIG 5** Principal-component analysis (PCA) is subjected to illustrate the dissimilarity of gut metabolomic profiling between healthy participants and enrolled diabetic CKD (d-CKD), hypertensive CKD (H-CKD), or CKD patients with no comorbidity (NC-CKD).

middle). The specific OTU-metabolite associations were shown to achieve convincing discrimination between healthy participants and CKD patients with ROC analyses. Utilization of the specific species-metabolite association discriminated NC-CKD, H-CKD, or d-CKD patients from healthy condition with an AUC value from 0.962, 0.901, or 0.913 (Fig. 8, right).

## DISCUSSION

A growing body of studies suggest the implication of gut dysbiosis and altered metabolite composition in the pathogenesis of CKD with underlying phenomenon, including mucosal permeability, systematic immunity, or metabolic disorders. With advancements in dual-omics approaches, identification of gut microbiota or metabolite profile allows the analysis of causation resulting in the deterioration of CKD. In this study, the specific gut dysbiosis, altered metabolite profile, and its association were characterized in the sub-population of CKD patients.

**TABLE 3** Statistics of gut metabolites with discriminative abundance in feces samples of CKD patients compared with healthy group

| Metabolite | KEGG | HMDB | Microbe | Hsa | CKD | VIP | *P* value | Fold change (CKD vs. healthy) | RSD (%) |
|---|---|---|---|---|---|---|---|---|---|
| Arachidic acid | C06425 | HMDB0002212 | Yes | Yes | NC | 3.04 | 0.017 | 24.20818 | 5.61 |
| N,N-Dimethylaniline | C02846 | HMDB0001020 | NA | NA | NC | 2.95 | 0.013 | 20.58342 | 8.57 |
| L-Phenylalanine | C00079 | HMDB0000159 | Yes | Yes | NC | 3.12 | 0.008 | 15.08339 | 9.04 |
| Dihomo-gamma-linolenic acid | C03242 | HMDB0002925 | Yes | Yes | NC | 2.54 | 0.023 | 10.98456 | 11.23 |
| N-Acetylputrescine | C02714 | HMDB0002064 | Yes | Yes | NC | 2.21 | 0.027 | 3.319421 | 15.48 |
| Hexamethylquercetagetin | NA | HMDB0029308 | NA | NA | NC | 1.97 | 0.035 | 0.422767 | 21.51 |
| 5,7,3′-Trihydroxy-4′-methoxyflavanone | NA | HMDB0030746 | NA | NA | NC | 2.04 | 0.026 | 0.411529 | 17.69 |
| 1,3,7-Trimethyluric acid | C16361 | HMDB0002123 | Yes | Yes | NC | 1.99 | 0.018 | 0.363065 | 13.44 |
| Homocarnosine | C00884 | HMDB0000745 | Yes | Yes | NC | 2.06 | 0.025 | 0.173692 | 11.27 |
| Epinephrine | C00788 | HMDB0000068 | Yes | Yes | NC | 2.87 | 0.018 | 0.171792 | 10.05 |
| Acetaminophen | C06804 | HMDB0001859 | NA | Yes | NC | 3.12 | 0.017 | 0.155547 | 11.34 |
| N-Acetylputrescine | C02714 | HMDB0002064 | Yes | Yes | NC | 3.55 | 0.024 | 0.108809 | 12.65 |
| Linoleic acid | C01595 | HMDB0000673 | Yes | Yes | NC | 2.57 | 0.015 | 0.089211 | 10.91 |
| Avocadyne | NA | HMDB0035473 | NA | NA | NC | 3.05 | 0.009 | 0.063397 | 8.77 |
| Ephedrine | C01575 | HMDB0015451 | Yes | Yes | NC | 3.11 | 0.011 | 0.052377 | 6.45 |
| Octadecylamine | NA | HMDB0029586 | NA | NA | NC | 2.85 | 0.018 | 0.026035 | 7.23 |
| Allocholic acid | C17737 | HMDB0000505 | Yes | NA | H-CKD | 3.581 | 0.0094 | 65.07163 | 3.41 |
| Aliskiren | NA | HMDB0015387 | NA | NA | H-CKD | 3.445 | 0.012 | 28.2064 | 7.25 |
| L-Glutamic acid | C00025 | HMDB0000148 | Yes | Yes | H-CKD | 3.17 | 0.017 | 24.53837 | 8.51 |
| Glycyrrhetinic acid | C02283 | HMDB0011628 | NA | NA | H-CKD | 3.04 | 0.021 | 17.47124 | 9.63 |
| Acetaminophen | C06804 | HMDB0001859 | NA | Yes | H-CKD | 2.85 | 0.014 | 16.6588 | 7.22 |
| Ondansetron | C07325 | HMDB0005035 | NA | NA | H-CKD | 2.71 | 0.023 | 7.111612 | 10.34 |
| Estrone glucuronide | C11133 | HMDB0004483 | NA | Yes | H-CKD | 2.34 | 0.027 | 6.045893 | 15.41 |
| Acamprosate | NA | HMDB0014797 | NA | NA | H-CKD | 2.25 | 0.031 | 4.777941 | 17.22 |
| Prednisone | C07370 | HMDB0014773 | NA | NA | H-CKD | 2.12 | 0.022 | 4.689358 | 13.41 |
| Salbutamol | C11770 | HMDB0001937 | NA | NA | H-CKD | 1.97 | 0.027 | 4.63981 | 16.22 |
| Spermine | C00750 | HMDB0001256 | Yes | Yes | H-CKD | 2.05 | 0.034 | 3.859989 | 23.41 |
| Hexamethylquercetagetin | NA | HMDB0029308 | NA | NA | H-CKD | 1.84 | 0.029 | 3.172898 | 27.45 |
| Mepivacaine | C07528 | C07528 | NA | NA | H-CKD | 1.88 | 0.033 | 2.659107 | 17.22 |
| Niflumic Acid | C13698 | HMDB0015573 | NA | NA | H-CKD | 1.84 | 0.042 | 0.440582 | 21.34 |
| Fumonisin A2 | NA | HMDB0034699 | NA | NA | H-CKD | 1.92 | 0.037 | 0.431233 | 19.55 |
| Zapotin | NA | HMDB0029461 | NA | NA | H-CKD | 1.77 | 0.025 | 0.424163 | 17.21 |
| Pregabalin | NA | HMDB0014375 | NA | NA | H-CKD | 2.01 | 0.023 | 0.330893 | 20.33 |
| Urobilinogen | C05791 | HMDB0004158 | Yes | Yes | H-CKD | 1.95 | 0.031 | 0.326445 | 21.42 |
| (S)-4′,5,7-Trihydroxy-6-prenylflavanone | C09832 | HMDB0037247 | NA | NA | H-CKD | 1.99 | 0.024 | 0.313458 | 18.57 |
| N1-Acetylspermine | C02567 | HMDB0001186 | NA | NA | H-CKD | 2.12 | 0.019 | 0.297418 | 17.24 |
| Fasciculic acid B | NA | HMDB0036438 | NA | NA | H-CKD | 2.42 | 0.021 | 0.162761 | 15.41 |
| Gefitinib | NA | HMDB0014462 | NA | NA | H-CKD | 2.28 | 0.013 | 0.051435 | 13.07 |
| Stearic acid | C01530 | HMDB0000827 | Yes | Yes | D-CKD | 3.14 | 0.012 | 19.091 | 11.34 |
| Glutaric acid | C00489 | HMDB0000661 | Yes | Yes | D-CKD | 2.94 | 0.017 | 5.721118 | 13.55 |
| Amiloride | C06821 | HMDB0014732 | NA | NA | D-CKD | 2.85 | 0.02 | 5.03763 | 9.81 |
| 2-Undecyl-4(1H)-quinolinone | NA | HMDB0032996 | NA | NA | D-CKD | 2.77 | 0.021 | 4.831606 | 14.52 |
| 2-Amino-3-methylimidazo[4,5-f]quinoline | C19180 | HMDB0029706 | NA | Yes | D-CKD | 2.51 | 0.019 | 3.087497 | 13.77 |
| 3,4-Dimethoxyphenylethylamine | NA | HMDB0041806 | NA | NA | D-CKD | 2.24 | 0.023 | 2.909365 | 15.63 |
| 5,7-Dihydroxyflavone | C10028 | HMDB0036619 | NA | NA | D-CKD | 1.87 | 0.031 | 2.025846 | 21.32 |
| L-Proline | C00148 | HMDB0000162 | Yes | Yes | D-CKD | 2.02 | 0.025 | 2.009057 | 22.51 |
| Hexamethylquercetagetin | NA | HMDB0029308 | NA | NA | D-CKD | 1.88 | 0.041 | 0.422767 | 27.33 |
| 5,7,3′-Trihydroxy-4′-methoxyflavanone | NA | HMDB0030746 | NA | NA | D-CKD | 1.74 | 0.038 | 0.411529 | 24.55 |
| 1,3,7-Trimethyluric acid | C16361 | HMDB0002123 | Yes | Yes | D-CKD | 2.15 | 0.031 | 0.363065 | 18.55 |
| Homocarnosine | C00884 | HMDB0000745 | Yes | Yes | D-CKD | 2.41 | 0.029 | 0.173692 | 17.21 |
| Epinephrine | C00788 | HMDB0000068 | Yes | Yes | D-CKD | 2.37 | 0.034 | 0.171792 | 16.03 |
| Acetaminophen | C06804 | HMDB0001859 | NA | Yes | D-CKD | 2.16 | 0.022 | 0.155547 | 18.45 |
| N-Acetylputrescine | C02714 | HMDB0002064 | Yes | Yes | D-CKD | 2.53 | 0.018 | 0.108809 | 13.57 |
| Linoleic acid | C01595 | HMDB0000673 | Yes | Yes | D-CKD | 2.77 | 0.021 | 0.089211 | 14.81 |
| Avocadyne | NA | HMDB0035473 | NA | NA | D-CKD | 2.63 | 0.019 | 0.063397 | 9.88 |
| Ephedrine | C01575 | HMDB0015451 | Yes | Yes | D-CKD | 3.04 | 0.015 | 0.052377 | 10.71 |
| Octadecylamine | NA | HMDB0029586 | NA | NA | D-CKD | 2.87 | 0.009 | 0.026035 | 8.54 |

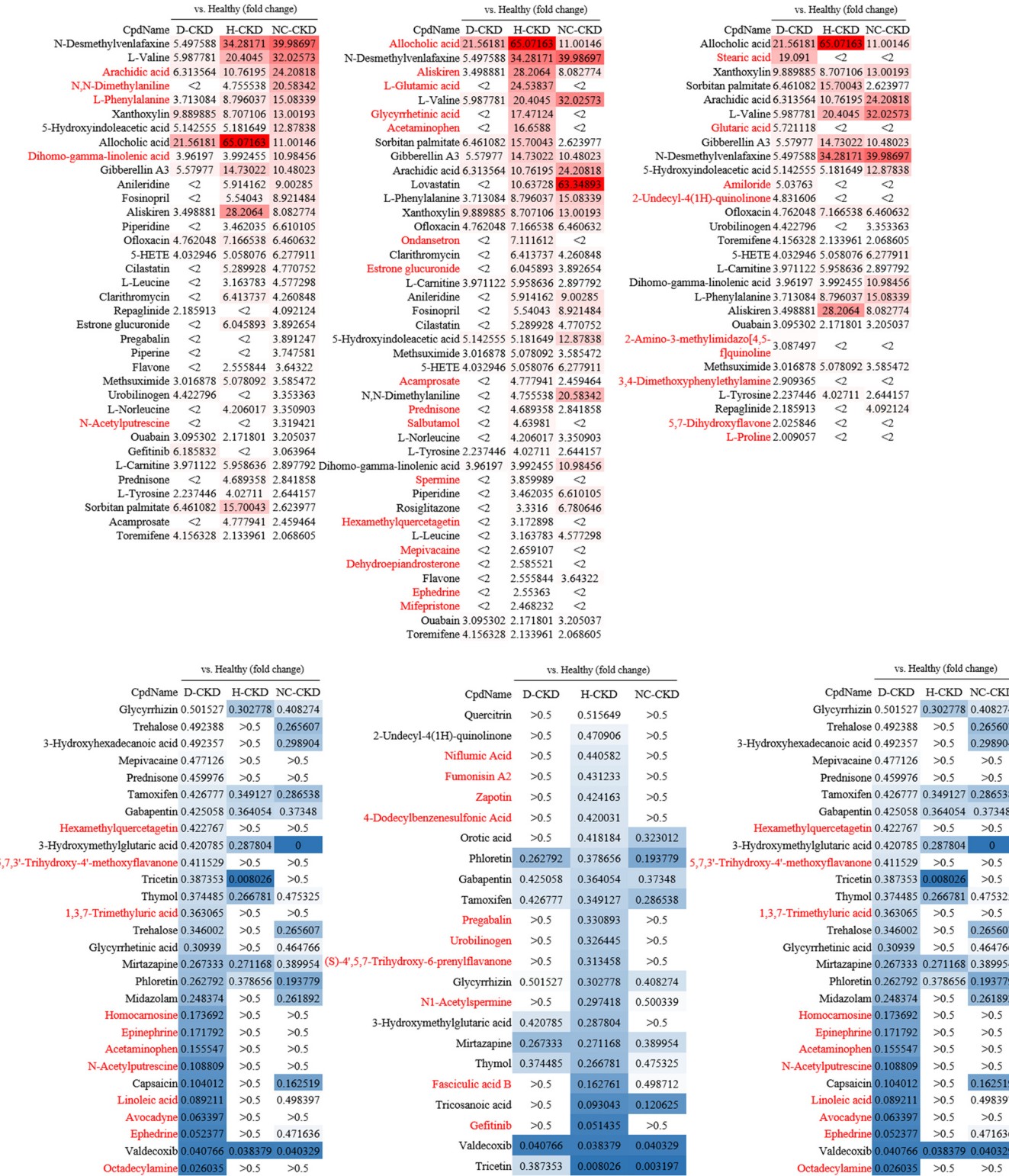

**FIG 6** Z-score heatmap is constructed with the statistically discriminative abundances of gut metabolites identified among enrolled diabetic CKD (d-CKD), hypertensive CKD (H-CKD), or CKD patients with no comorbidity (NC-CKD). Relevance between the discriminative increases (upper panel) or decreases (lower panel) in the identified metabolites and recruited CKD patients is shown in a heat map chart (red character). Significance of listed metabolites was evaluated using the value of variable importance in projection value (VIP) and alteration in relative abundance from pairwise PLD-DA analysis and Wilcoxon rank-sum test, with VIP > 1.5, alteration in relative abundance (-2> fold change >2), and *P* value < 0.05 as the cut-off for significance.

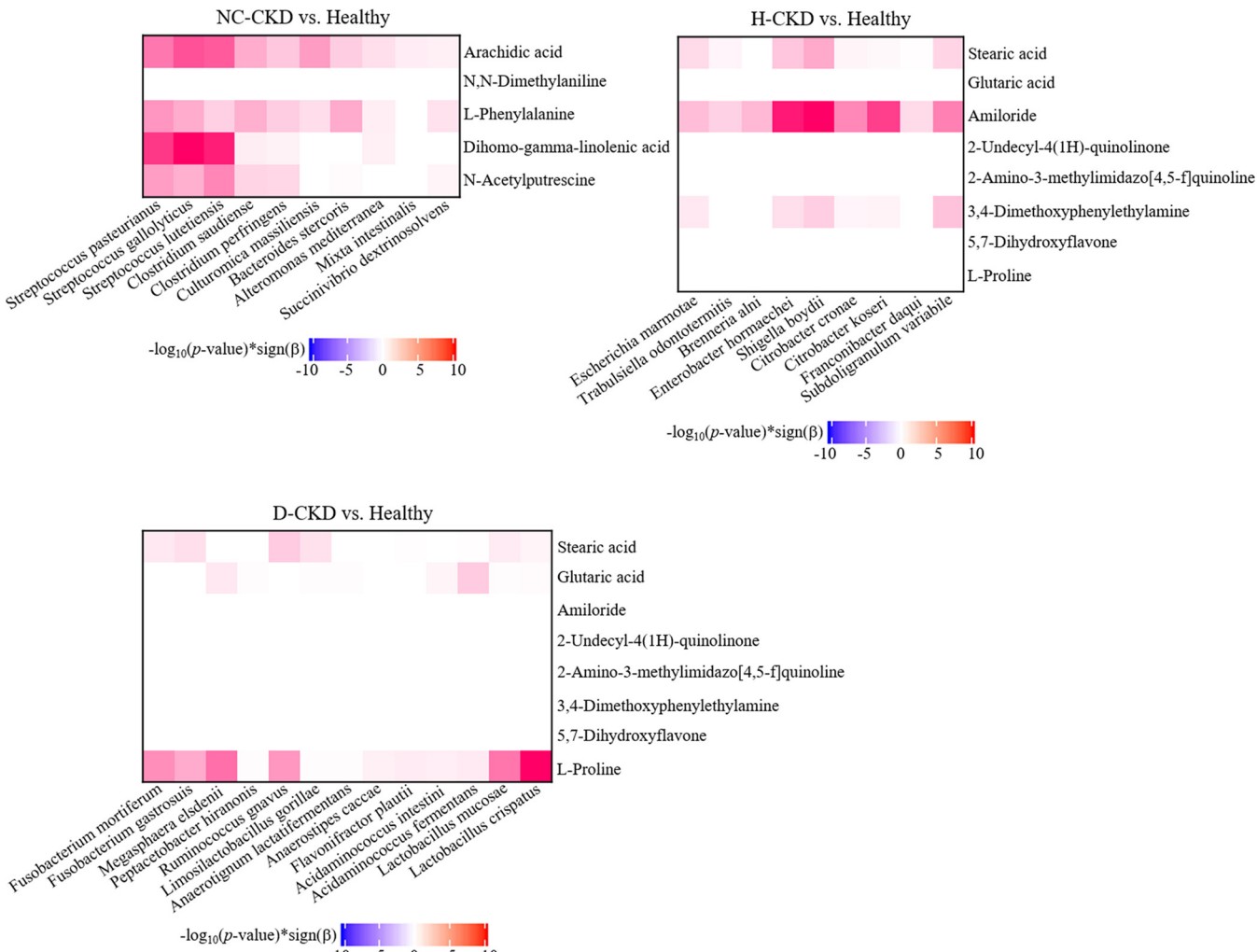

**FIG 7** Associations among CKD-enriched gut metabolites and identified OTUs in enrolled CKD patients. The association between identified metabolites and OTUs along with CKD occurrence is shown in a Heatmap. The metabolites-OTUs associations were evaluated with the utilization of zero-inflated negative binomial (ZINB) regressions. The strengths of associations were measured by -log$_{10}$(P-value)*sign(Beta) from the results of ZINB regressions and P value < 0.05 was identified as the cut-off for significance.

Hypertension is considered a well-known and critical risk factor toward CKD occurrence, and the related cohort study documented the association between hypertension and approximately 90% of patients diagnosed with stage 3 to stage 5 CKD (26). Gut dysbiosis was identified in hypertensive animal model or patients diagnosed with hypertension (27, 28). In brief, increases in the opportunistic pathogens, including *Enterococcus*, *Enterobacteriaceae*, and *Klebsiella* genera were identified under the hypertensive condition (29, 30). In contrast, the medical intervention toward gut dysbiosis, such as supplementation with high-fiber diet or probiotics, was demonstrated to exhibit blood pressure-lowering effect in the animal model or clinical patients (31, 32). In addition to hypertension, gut dysbiosis was characterized as well in patients diagnosed with diabetic kidney disease (DKD) which is a critical cause of ESRD (33). Recent studies disclosed the relevance of gut dysbiosis in richness or diversity with the occurrence of DKD (34, 35). Previous studies demonstrated a reduction in the butyrate-producing species, such as *Roseburia intestinals* or *Faecalibacterium prausnitzii*, with a concomitant increase in the opportunistic pathogen, including *Bacteroides*, *Clostridium*, *Eggerthella*, and *Escherichia* genera in the T2DM patients (36). The close relevance between decline in abundance of the butyrate-producers and glycemic condition with anti-diabetic intervention was documented, which suggest the application of microbial signature on evaluating the severity of DM or DKD (37). The

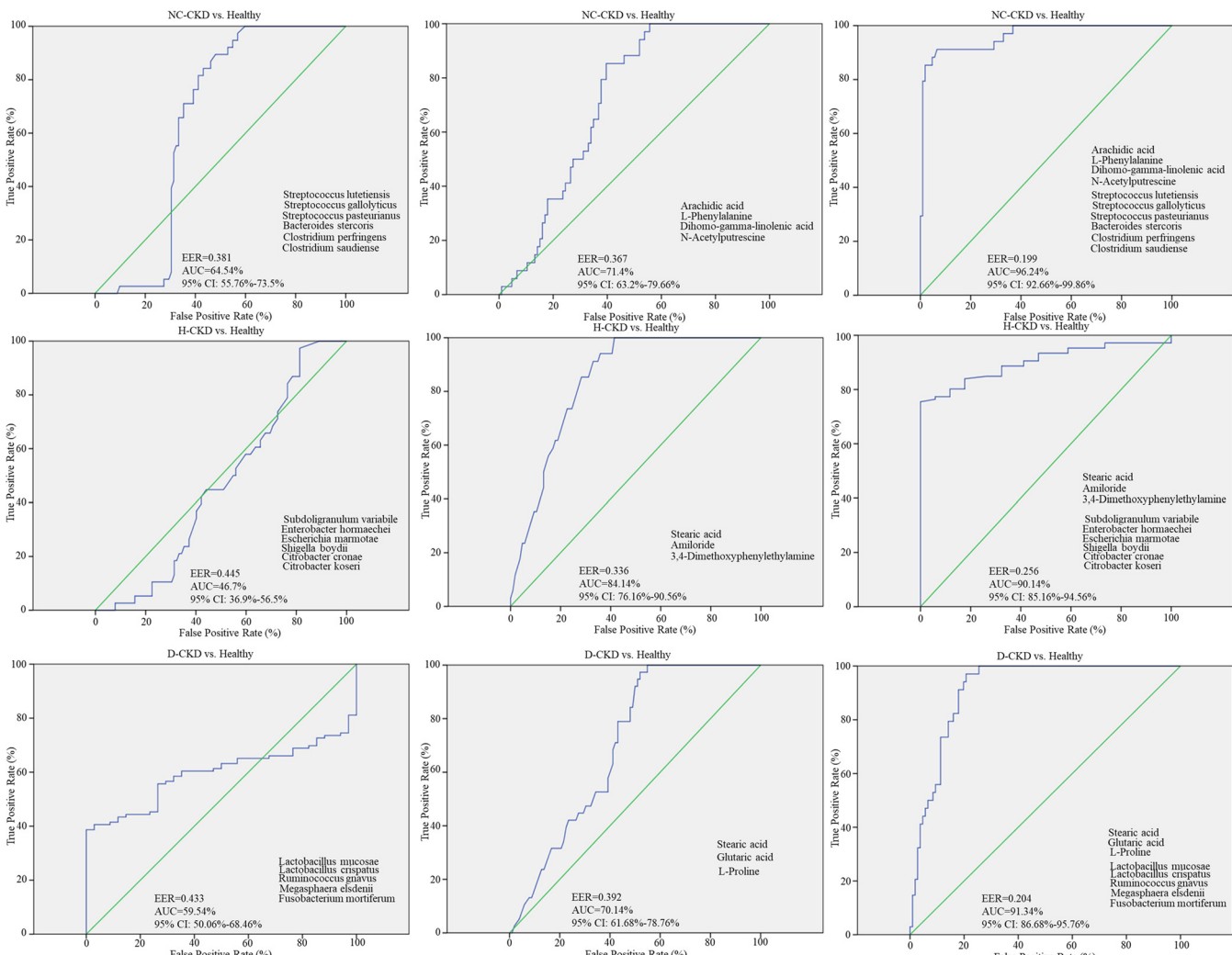

**FIG 8** Predictive performance of gut microbiota or metabolite toward the occurrence of CKD with distinct pathogenic factor was evaluated using the random forests model. The area under the receiver operating characteristics (ROC) curve (AUC) was utilized to distinguish the enrolled diabetic CKD (d-CKD), hypertensive CKD (H-CKD), or CKD patients with no comorbidity (NC-CKD), from the healthy participants with the discriminative abundances of identified OTUs (left), the changes in intensity of gut metabolites (middle), or strength of OTUs-metabolites association (right).

relatively high abundances of *Oxalobacter Formigenes* and *Veillonella Parvula* were relevant to the bile acid metabolism and identified in gut microbiota of H-CKD patients compared to those of healthy counterparts, which potentially exhibited predictive utility at the occurrence or early stage of H-CKD (38). Similarly, the relatively abundant levels of four lipopolysaccharide producing Gram-negative phyla, including *Bacteroidetes*, *Proteobacteria*, *Verrucomicrobia* and *Fusobacteria* were determined in the gut microbiota of the T2DM-CKD compared with healthy groups (39). Nevertheless, the species-level resolution toward identification of d-CKD-related microorganisms and predictive utility of these candidate toward the occurrence of d-CKD should be further determined.

Trimethylamine N-oxide (TMAO) is an amine compound generated by gut microbial community with the carnitine-rich food and subsequently oxidized by liver-produced flavin monooxygenase 3 (40). The correlation between high serum level of TMAO and hypertension has been widely proven a major and critical causative factor involved in hypertensive CKD (41). In our study, the relatively high abundance of carnitine and TMAO was commonly characterized among all CKD patients, which suggested the impact of high TMAO on occurrence of CKD with distinct comorbidity. In contrast, the beneficial short chain fatty acid (SCFA) is produced by gut bacteria with the metabolism of fiber-rich food, which exerted protective actions against inflammation, metabolic

disorders, or dysfunction in mice or rat models (42, 43). Analytic results of microbial communities suggested that supplementation of high-fiber diets manipulated the DM-associated dysbiosis with the reduced growth of the pathobiont, such as urease-produced *Bilophila wadsworthia*, and elevated abundances of SCFA-producing probiotics, including *Prevotella* and *Bifidobacterium* in the streptozotocin-induced diabetes model (44). Among the derived SCFAs, butyrate has been widely reported to exhibit defensive impact against oxidative stress, strengthen mucosal permeability, or hamper nephropathy progression through epigenetic regulation, subsequently accounting for renal dysfunction in DKD (45, 46). In this study, the increases in relative abundances of bile acid and saturated fatty acid, such as allocholic acid or stearic acid, were characterized among all CKD patients or specifically identified in d-CKD patients, which suggested the close relevance between dietary intake and severity or progression of d-CKD.

Alteration in gut microbiota has been considered an important factor in manipulating the production of toxins involved in CKD-associated complications (47), which may function as the disease-specific biomarker as well. In our study, the discriminative gut microbiota-metabolite association was demonstrated to exert the most specific utility to differentiate CKD patient from healthy group. In addition, the influence of altered metabolite profile on renal function through particular pathway suggested its application for mechanistic investigation or clinical intervention. The gut microbiota or metabolite profile is diverse in different region or race with the dietary intake, life style, or study pipeline. Therefore, this distinction is critical and worthy of further pursue for the development of personal medicine toward CKD patient.

**Conclusion.** In this study, utilization of dual-omics approach provides high-resolution gut dysbiosis and altered metabolite profiles which could function as a comorbidity-associated marker for the early prevention, screening, or design of medical intervention to the CKD patient or high-risk population. Prior to the potential achievement, a longitudinal and subsequent functional study is essential for better understanding of the causative effect of altered gut microbiota or metabolite on the pathogenesis of CKD associated with distinct risk factor.

## MATERIALS AND METHODS

**Ethics statement of sample collection.** Enrollment of clinical participants, fecal sample collection, and the dual-omics assays were conducted according to the guidelines of the Declaration of Helsinki, which has been approved by the Institutional Review Board of Taipei Medical University (approval no. N202003133). Patients diagnosed with CKD were recruited from the Division of Nephrology at Taipei Municipal WanFang Hospital, and healthy participants were enrolled from the Health Examination Center at Taipei Municipal WanFang Hospital. CKD was defined in accordance with 2002 clinical practice guidelines (48). Comprehensive physical and biochemistry examination were conducted on all participants to ensure related condition with respect to kidney function, diabetes, and hypertension. A standard questionnaire was applied to evaluate lifestyle of all participants.

**Extraction of total genomic DNA in feces.** Feces sample was *in vitro* collected and stored using DNA/RNA Shield Fecal Collection tubes (Zymo Research, Irvine, CA, USA) to diminish environmental disturbance. Total genomic DNA was prepared using a Quick-DNA Fecal/Soil Microbe Microprep Kit (Zymo Research, Irvine, CA, USA) according to the manufacturer's instructions. Prior to functional analysis, the extracted DNA sample was quantified using a fluorometric assay (GeneCopoeia, Rockville, MD, USA).

**16S ribosomal (r)RNA gene sequencing.** Gut microbiota was identified using a long-read sequencing platform. In brief, 10 ng total gDNA was subjected for library construction of 16S ribosomal (r)RNA gene using the SQK-16S024 Barcoding kit (Oxford Nanopore Technologies (ONT), Oxford, UK) according to the manufacturer's protocol. The barcoded *16S rRNA* gene was washed and eluted from the magnetic beads (AMPure XP, Beckman Coulter, High Wycombe, UK). Pooled library containing 2 ng barcoded DNA of each participant was ligated with the adapter and sequenced on MinION flow cells (FLO-MIN106D R9.4.1; MinION instrument; ONT). The average read number of each sample was 100,000 to meet an even and sufficient reading depth toward identification of gut microbial community.

**Processing, annotation, and statistical analysis of sequencing results.** The quality check and clustering of OTU from raw reads was performed using CLC genomics workbench (Qiagen v22.0.2; CLC bio, Aarhus, Denmark) and Microbial Genomics Module v22.1 (Qiagen). Low depth samples ($<$ 10,000 reads per sample) and low read number ($<$ 4,000 reads per output file) were excluded from the analysis. Default parameters were used for identification and removal of chimeric reads using CLC genomics workbench. Qualified reads were mapped to 20,959 complete *16S rRNA* gene reference sequence curated from the Bacterial 16S rRNA RefSeq Targeted Loci Project (Accession No. PRJNA33175, NCBI) with 97% similarity by using Minimap2 program. The OUT table was subjected to the construction of phylogenetic tree using MUSCLE 2.0 and Maximum Likelihood Phylogeny tools available at CLC Genomics

Workbench. Alpha diversity metrics (Simpson and Shannon indices) was estimated by using the phyloseq R package based on rarefed OTU table. The weighted UniFrac or Bray-Curtis indices was defined with the pipeline to indicate the inter-sample dissimilarity (beta diversity). Differential abundance of identified taxa between each group was synchronously assessed using the linear discriminant analysis (LDA) effect size (LEfSe) method with default settings on the website algorithm (https://huttenhower.sph.harvard.edu/galaxy/root). The relative difference of identified taxa was identified significantly discriminative with a $P$ value $< 0.05$ and an LDA score ($\log_{10}$) $> 3$ or $< -3$.

**Extraction of fecal metabolites.** The extraction of fecal metabolites was commissioned to a commercial company (BIOTOOLS Co., Ltd.; Taipei, Taiwan). In brief, 50 mg feces sample was mixed in 1 mL extract solution (acetonitrile: methanol: water = 2: 2: 1) with vigorous vortex. The mixture was homogenized, sonicated, and incubated at $-20$℃ for 1 h. The supernatant was separated with centrifugation (12000 rpm for 15 min at 4℃) and then transferred to a glass vial.

**Untargeted metabolomics analysis.** In this study, the untargeted identification of fecal metabolites was commissioned to a commercial company (BIOTOOLS Co., Ltd.; Taipei, Taiwan). In brief, 10 $\mu$L of extract prepared from each sample was applied to a vanquish focused ultra-performance liquid chromatography (UPLC) coupled with an Orbitrap Elite Mass Spectrometry (Thermo Fisher Scientific; San Jose, CA, USA). The binary mobile phase was composed of deionized water containing 0.1% formic acid (solvent A) and LC-MS grade acetonitrile with 0.1% formic acid (solvent B). Throughout the linear gradient elution, the percentage of Solvent B was linearly increased from 5% to 100% for 7 min, kept constant for 3 min, then decreased to 5% in 1 min. Blank injection was applied to diminish carry-over and QC application was conducted to normalize the peak area. Mass spectrometry data were harvested in positive mode with a default data-dependent acquisition set. One MS full scan was applied in a profile mode at 60000 resolution, followed by 10 data-dependent MS2 scans at 15000 resolution. The mass scan range was set from 70 to 1000 $m/z$ and the normalized collision energy (NCE) was set to 25. The spray voltage, and the capillary temperature, the sheath gas and the aux gas was applied with the default set.

**Processing, annotation, and statistical analysis of UPLC-MS/MS data.** The raw data was converted to the mzXML format using ProteoWizard software for following analysis. The converted results were processed with an in-house program based on XCMS using R program for peak detection, extraction, alignment, and integration (BIOTOOLS Co., Ltd.; Taipei, Taiwan). An in-house MS2 data-base (BiotreeDB; BIOTOOLS Co., Ltd.; Taipei, Taiwan) was applied for the annotation of metabolomic profile, of which the cut-off was set to 0.3.

**Statistical analysis.** Statistic results to the generated data were shown as the mean $\pm$ standard error (SEM). Continuous variables of this study is evaluated using a one-way analysis of variance (ANOVA) coupled with Tukey's multiple comparison *post hoc* test. A variable was identified significant with a $P$ value of $< 0.05$ (*, $P < 0.05$; **, $P < 0.01$; ***, $P < 0.005$). The species-metabolite association under specific condition was assessed using zero-inflated negative binomial (ZINB) regression (R package pscl). The read number of identified species was defined as a dependent variable and the strength of identified metabolite was defined as an independent variable in ZINB regressions. The association was shown by -$\log_{10}(P$-value)* sign ($\beta$), of which $\beta$ presented the regression of the metabolite. The utility of species, metabolite, or its association for predicting the diagnosis of CKD was evaluated with the results of the receiver operating characteristic (ROC) curve and area under the ROC curve (AUC) ratio generated by using SPSS Statistics 19 (IBM, Armonk, NY). The relevance between OTU number and equal error rate (EER) was evaluated by using the built-in RFCV function of the randomForest package (R version. 4.6–14). A cross-validation step was subjected to minimize the EER with an optimal predictor number in this study (49).

**Data availability.** Raw *16S rRNA* sequencing data sets were deposited with the NCBI BioProject database under accession number PRJNA899930.

## ACKNOWLEDGMENTS

This work was supported by a grant (MOST 110-2320-B-038-059) from the Ministry of Science and Technology, Taiwan; (110-6603-005-400) from the Ministry of Education, Taiwan.

We declare that no conflict of interest existed.

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
