## [Reviewer comments · Microbiology Spectrum]

Microbiology Spectrum

Exploring the relevance between gut microenvironment and chronic kidney disease with distinct pathogenic factor

Tso-Hsiao Chen, Chung-Yi Cheng, Chun-Kai Huang, Yi-Hsien Ho, and Jung-Chun Lin

Corresponding Author(s): Jung-Chun Lin, Taipei Medical University

Review Timeline:

Submission Date:	July 21, 2022
Editorial Decision:	October 7, 2022
Revision Received:	October 20, 2022
Editorial Decision:	November 9, 2022
Revision Received:	November 10, 2022
Accepted:	November 22, 2022

Editor: Wei-Hua Chen

Reviewer(s): Disclosure of reviewer identity is with reference to reviewer comments included in decision letter(s). The following individuals involved in review of your submission have agreed to reveal their identity: Na Fei (Reviewer #1)

Transaction Report:

DOI: <https://doi.org/10.1128/spectrum.02805-22>

October 7, 2022

Prof. Jung-Chun Lin
Taipei Medical University
School of Medical Laboratory Science and Biotechnology, College of Medical Science and Technology
250 Wu-Hsing Street
Taipei 110
Taiwan

Re: Spectrum02805-22 (Exploring the relevance between gut microenvironment and chronic kidney disease with distinct pathogenic factor)

Dear Prof. Jung-Chun Lin:

Thank you for submitting your manuscript to Microbiology Spectrum. Your manuscript has been evaluated by two external experts. Although they found the study interesting, they also raised substantial concerns. I therefore would like to invite you to submit a revised version of your manuscript to properly address the issues in details.

Link Not Available

Sincerely,

Wei-Hua Chen

Journals Department
Reviewer comments:

Reviewer #1 (Comments for the Author):

Review report

The manuscript by Tso-Hsiao Chen and colleagues addresses an exciting area of the presence of resident microbiota in the development of Tso-Hsiao Chen (CRD). Liquid chromatography coupled-mass spectrometry and long-read sequencing were applied to identify gut metabolites and microbiome with statistically discriminative abundance in diabetic CKD patients (n=39),

hypertensive CKD patients, or CKD patients without comorbidity as compared to those of healthy participants (n=60). The association between CKD-related species and metabolites was evaluated using zero-inflated negative binomial (ZINB) regression. Furthermore, the predictive utility of identified OTU, metabolite, or species-metabolite association toward the diagnosis of incident chronic kidney disease with distinct pathogenic factors was assessed using the random forest regression model and the receiver operating characteristic (ROC) curve. In this study, the specific gut dysbiosis, altered metabolite profile, and its association were characterized in the sub-population of CKD patients. The study's shortcoming is that the clinical importance of the study, especially the predictive function of the dysbiosis to disease, is not comprehensive. Here are my comments:

Major comments:

- 1, "gut microenvironment" means the immediate small-scale or a part of the gut's environment, especially as a distinct part of a larger environment. It's not specific to the microbiome. However, this article studies the predictive function of microbial structure and metabolites in CKD. It might be more precise to use another relative term.
- 2, In the Weighted Unifrac analysis (figure 2): pairwise comparison can show the difference in the beta-diversity between groups is significant or not. Otherwise, it's hard to conclude that "These results indicated gut dysbiosis in all CKD patients as compared to those of healthy participants." from the description.
- 3, Line 178: The Conclusion "These results suggested the association between gut dysbiosis and discriminative disease condition involved in the pathogenesis of CKD." need to be modified to structural changes of microbiota involved in the pathogenesis of CKD before the authors prove these changes related to dysbiosis.
- 4, The predictive function of gut microbial characters in CKD is not discussed in the discussion section. However, using microbial features to predict disease is an important application. The authors also need to compare with other studies using microbes to predict the development of diseases. Besides, the association of the microbial characters was not discussed either.
- 5, Raw data need to be deposited in a public repository. "The raw 16S rRNA sequencing data for demonstrating the results of this article is not submitted to any public database."
- 6, Line 349: The average read number of each sample was 40,000 to meet a reading depth of 50. Why the reading depth was set as 50, which is critically low? What is the range of reads numbers for all the samples?
- 7, Line 133: In this study, the genomic DNA extracted from the fecal sample was subjected to the long-read sequencing platform (MinION, ONT, Oxford, UK) to characterize the gut microbial communities in enrolled participants.
- 8, What is long sequences? There's no description for the target's description in the manuscript.
- 9, Quality control of the raw data was not described in the manuscript.
- 10, Methods for Alpha diversity and beta diversity were not described in the manuscript.

Minor comments:

- 1, There are several typos and grammar mistakes.
- 2, 16S rRNA needs to be changed to 16S rRNA gene

Reviewer #2 (Comments for the Author):

Chen et al. studied the gut microbiome and metabolite profile from patients diagnosed with hypertensive CKD (H-CKD), diabetic CKD (D-CKD), CKD patients with no comorbidity condition (NC-CKD), and healthy participants (HP). ONT long-read sequencing and LC-QTOFMS/MS techniques were applied to identify microbes and metabolites with statistical procedures including a machine learning method. This study has the potential importance of identifying pathogenesis-associated markers across healthy participants and high-risk CKT populations toward the early screening, prevention, diagnosis, or personalized treatment of CKD, but the manuscript should be significantly updated to be published.

Major points:

1. I am not a native English reviewer, but I have found too many inappropriate English grammar and typos. English correction is essential. A few examples that I found were noted in the minor points, but those are only parts of them.
2. The authors stated that "Data availability: The raw 16S rRNA sequencing data for demonstrating the results of this article is not submitted to any public database." I am wondering why the data were not deposited. The reviewer strongly recommends authors deposit the data into a publicly available repository for researchers to reproduce the results. An open dataset will be very beneficial to other researchers to validate the results and study other factors. If there is a reason to not disclose the data, please explain the reason as "The data that support the findings of this study are not openly available due to [reasons of sensitivity e.g. human data] and are available from the corresponding author upon reasonable request [include information on the data's location, e.g. in a controlled access repository where relevant]."

3. The authors stated that "The utility of species, metabolite, or its association for predicting the diagnosis of CKD was evaluated with the results of the receiver operating characteristic (ROC) curve and area under the ROC curve (AUC) ratio generated by using SPSS Statistics." and those markers were highlighted in Figures 6 and 7, but it would be great to provide a table (or tables) of them for reads to check those markers easily.

4. Predictive performance of gut microenvironment toward the occurrence of CKD with distinct pathogenic factors was evaluated using the random forests model and AUC curve, but the model training dataset and independent dataset were not split, and it is very highly overfitted on the training model. This result and section should be re-evaluated and re-stated with the split datasets. Cross-validation also should be done to validate the model accuracy and model bias. Not only the AUC curve but also the error rate is also a good metric to show. Also, Figures 1-7 have each figure number in the figure, but not in Figure 8. The method section for this Random Forest must be rewritten.

5. The authors stated that "Qualified reads were mapped to the 16S rRNA reference released from the NCBI database." What NCBI database was used? Complete microbial genomes? From RefSeq? Including draft genomes? The database and mapping method should be written clearly.

6. In the metabolites analysis section, the authors stated that "The converted results were processed with an in-house program based on XCMS using R program for peak detection, extraction, alignment, and integration." Is the in-house program made by authors or not? Publicly available? Again, the manuscript needs to provide all details about such methods and software tools for readers to understand the procedure of the analysis and to be convinced by the provided results.

Minor points:

Please go through the manuscript and polish English and grammar. For example,

* Page 3: Line 3: "study"  "studies"

* Page 3: Lines 6-7: "a liquid chromatography coupled-mass spectrometry and long-read sequencing was"  "a liquid chromatography coupled-mass spectrometry and long-read sequencing were"

* Page 5: Line 9: delete "in"

* Page 5: Line 12-13: change "or" to "and" or "have" to "has"

* Page 5: Line 18: "complication"  "complications"

* Page 6: Line 4-5: "With proliferation of these pathogenic bacteria come production of uremic toxins"  "With the proliferation of these pathogenic bacteria comes the production of uremic toxins"

...

* Page 21: Line 3: "The quantity 353 of sequencing reads was evaluated t using Microbial Genomics Module"  delete "t"

* Page 21: Line 9: "algorism"  "algorithm"

* Pag3 23: Line 1: "sofeware"  "software"

Staff Comments:

Preparing Revision Guidelines

Please return the manuscript within 60 days; if you cannot complete the modification within this time period, please contact me. If you do not wish to modify the manuscript and prefer to submit it to another journal, please notify me of your decision immediately so that the manuscript may be formally withdrawn from consideration by Microbiology Spectrum.

If your manuscript is accepted for publication, you will be contacted separately about payment when the proofs are issued; please follow the instructions in that e-mail. Arrangements for payment must be made before your article is published. For a

complete list of **Publication Fees**, including supplemental material costs, please visit our website.

Response to Reviewers' Comments
(Manuscript Number: Spectrum02805-22)

Dear Editor:

We thank you for your response and for allowing revision of our manuscript (Spectrum02805-22; *Exploring the relevance between gut microenvironment and chronic kidney disease with distinct pathogenic factor*). The manuscript was revised in line with the valuable suggestions and comments of the reviewer. We hope that the revised manuscript achieves reviewer satisfaction. Our point-by-point responses to all specific reviewer comments, suggestions, and queries are as follows.

Response to Reviewer 1's comments

The manuscript by Tso-Hsiao Chen and colleagues addresses an exciting area of the presence of resident microbiota in the development of Tso-Hsiao Chen (CRD). Liquid chromatography coupled-mass spectrometry and long-read sequencing were applied to identify gut metabolites and microbiome with statistically discriminative abundance in diabetic CKD patients (n=39), hypertensive CKD patients, or CKD patients without comorbidity as compared to those of healthy participants (n=60). The association between CKD-related species and metabolites was evaluated using zero-inflated negative binomial (ZINB) regression. Furthermore, the predictive utility of identified OTU, metabolite, or species-metabolite association toward the diagnosis of incident chronic kidney disease with distinct pathogenic factors was assessed using the random forest regression model and the receiver operating characteristic (ROC) curve. In this study, the specific gut dysbiosis, altered metabolite profile, and its association were characterized in the sub-population of CKD patients. The study's shortcoming is that the clinical importance of the study, especially the predictive function of the dysbiosis to disease, is not comprehensive. Here are my comments:

1. "gut microenvironment" means the immediate small-scale or a part of the gut's environment, especially as a distinct part of a larger environment. It's not specific to the microbiome. However, this article studies the predictive function of microbial structure and metabolites in CKD. It might be more precise to use another relative term.

Response:

We appreciate and agree with the reviewer's suggestion. This term was revised to "gut microbiota and metabolites" throughout the revised manuscript.

2. In the Weighted Unifrac analysis (figure 2): pairwise comparison can show the difference in the beta-diversity between groups is significant or not. Otherwise, it's hard to conclude that "These results indicated gut dysbiosis in all CKD patients as compared to those of healthy participants. " from the description.

Response:

The sentence was revised to tone down the description toward the results of statistical analyses in this study (**page 9, lines 10-18**).

3. Line 178: The Conclusion "These results suggested the association between gut dysbiosis and discriminative disease condition involved in the pathogenesis of CKD. " need to be modified to structural changes of microbiota involved in the pathogenesis of CKD before the authors prove these changes related to dysbiosis.

Response:

The sentence was revised according to the reviewer's suggestion (**page 11, lines 3-5**).

4. The predictive function of gut microbial characters in CKD is not discussed in the discussion section. However, using microbial features to predict disease is an

important application. The authors also need to compare with other studies using microbes to predict the development of diseases. Besides, the association of the microbial characters was not discussed either.

Response:

The **Discussion** section was comprehensively revised to meet the reviewer's comment (**page 16, lines 9-19; page 18, lines 4-13**).

5. Raw data need to be deposited in a public repository. "The raw 16S rRNA sequencing data for demonstrating the results of this article is not submitted to any public database. "

Response:

We agree with the reviewer's suggestion regarding the data availability. The raw 16S rRNA sequencing data is available from the corresponding author with reasonable purpose (**page 26, lines 1-2**).

6. Line 349: The average read number of each sample was 40,000 to meet a reading depth of 50. Why the reading depth was set as 50, which is critically low? What is the range of reads numbers for all the samples?

Response:

We apologize for the mistake in the **Materials and methods** section. The qualified read number of each sample was consistently set as 100,000 throughout our previous (**Ref. 25**) and present study to meet the even and sufficient reading depth, and low-depth sample (with <10,000 reads) was omitted from the analysis (**page 22, lines 3-4**). Qualified read numbers ranging from 100,000 to 120,000 reads for each sample was subjected to subsequent analyses as described in the **Materials and methods** section (**page 21, lines 13-15**).

7. Line 133: In this study, the genomic DNA extracted from the fecal sample was subjected to the long-read sequencing platform (MinION, ONT, Oxford, UK) to characterize the gut microbial communities in enrolled participants.

Response:

The sentence was revised according to the reviewer's suggestion (**page 8, lines 13-15**).

8. What is long sequences? There's no description for the target's description in the manuscript.

Response:

We are sorry that we do not find out the term "long sequences" throughout the manuscript. Nevertheless, we provided the introduction regarding "long-read sequencing" in the **Introduction** section of the revised manuscript (**page 6, lines 17-19; page 7, lines 1-2**).

9. Quality control of the raw data was not described in the manuscript.

Response:

The analytic pipeline toward quality control of the raw reads was further described in the **Materials and methods** section of revised manuscript (**page 22, lines 1-6**).

10. Methods for Alpha diversity and beta diversity were not described in the manuscript.

Response:

The analytic pipeline toward alpha and beta diversity of gut microbiota was further described in the **Materials and methods** section of revised manuscript (**page 22, lines 6-14**).

Minor comments:

1. There are several typos and grammar mistakes.

Response:

The manuscript has been checked and edited by native English editor to eliminate typo and grammatical error prior to the revision.

2. 16S rRNA needs to be changed to 16S rRNA gene

Response:

This term has been revised throughout the manuscript.

Response to Reviewer 2's comments

Chen et al. studied the gut microbiome and metabolite profile from patients diagnosed with hypertensive CKD (H-CKD), diabetic CKD (D-CKD), CKD patients with no comorbidity condition (NC-CKD), and healthy participants (HP). ONT long-read sequencing and LC-QTOFMS/MS techniques were applied to identify microbes and metabolites with statistical procedures including a machine learning method. This study has the potential importance of identifying pathogenesis-associated markers across healthy participants and high-risk CKT populations toward the early screening,

prevention, diagnosis, or personalized treatment of CKD, but the manuscript should be significantly updated to be published.

1. I am not a native English reviewer, but I have found too many inappropriate English grammar and typos. English correction is essential. A few examples that I found were noted in the minor points, but those are only parts of them.

Response:

The manuscript has been checked and edited by native English editor to eliminate typo and grammatical error prior to the revision.

2. The authors stated that "Data availability: The raw 16S rRNA sequencing data for demonstrating the results of this article is not submitted to any public database." I am wondering why the data were not deposited. The reviewer strongly recommends authors deposit the data into a publicly available repository for researchers to reproduce the results. An open dataset will be very beneficial to other researchers to validate the results and study other factors. If there is a reason to not disclose the data, please explain the reason as "The data that support the findings of this study are not openly available due to [reasons of sensitivity e.g. human data] and are available from the corresponding author upon reasonable request [include information on the data's location, e.g. in a controlled access repository where relevant]."

Response:

We appreciate and agree with the reviewer's suggestion regarding the data availability. The raw 16S rRNA sequencing data is available from the corresponding author with reasonable purpose (**page 26, lines 1-2**).

3. The authors stated that "The utility of species, metabolite, or its association for predicting the diagnosis of CKD was evaluated with the results of the receiver operating characteristic (ROC) curve and area under the ROC curve (AUC) ratio generated by using SPSS Statistics." and those markers were highlighted in Figures 6 and 7, but it would be great to provide a table (or tables) of them for reads to check those markers easily.

Response:

The results presented in Figure 6 and 7 was summarized in **Table 3** according to the reviewer's suggestion.

4. Predictive performance of gut microenvironment toward the occurrence of CKD with distinct pathogenic factors was evaluated using the random forests model and AUC curve, but the model training dataset and independent dataset were not split, and it is very highly overfitted on the training model. This result and section should be re-evaluated and re-stated with the split datasets. Cross-validation also should be done to validate the model accuracy and model bias. Not only the AUC curve but also the error rate is also a good metric to show. Also, Figures 1-7 have each figure number in the figure, but not in Figure 8. The method section for this Random Forest must be rewritten.

Response:

1. We agree with the reviewer's suggestion and apologize for the mistake on describing the related results. The utility of characterized species or metabolites on distinguishing CKD patient with distinct comorbidity from healthy group was evaluated with the distinct participants that were enrolled in our previous study (**Ref. 25**). The equal error rate (EER) of random forests model throughout this cross-

validation was provided in Figure 8. The results and **Materials and Methods** section were subsequently revised (**page 25, lines 13-17**).

2. The number of Figure 8 was provided in the revised figure.

5. The authors stated that "Qualified reads were mapped to the 16S rRNA reference released from the NCBI database." What NCBI database was used? Complete microbial genomes? From RefSeq? Including draft genomes? The database and mapping method should be written clearly.

Response:

The qualified reads generated in this study were mapped to 20,959 complete 16S rRNA reference that curated from the Bacterial 16S Ribosomal RNA RefSeq Targeted Loci Project submitted to NCBI (Accession No. PRJNA33175, NCBI) by using Minimap2 program. The origin of reference sequence and mapping program was further described in the **Materials and methods** section (**page 22, lines 6-9**).

6. In the metabolites analysis section, the authors stated that "The converted results were processed with an in-house program based on XCMS using R program for peak detection, extraction, alignment, and integration." Is the in-house program made by authors or not? Publicly available? Again, the manuscript needs to provide all details about such methods and software tools for readers to understand the procedure of the analysis and to be convinced by the provided results.

Response:

In this study, we commissioned a commercial company (BIOTOOLS Co., Ltd.; Taipei, Taiwan) to execute the untargeted identification of gut metabolites and following analyses, including peak detection, extraction, alignment, and integration. The in-house program is constructed by the commercial company (BIOTOOLS Co.,

Ltd.; Taipei, Taiwan), of which the publicity is not available. The detailed information was provided in the revised **Materials and methods** section (**page 23, lines 3-4; lines 11-12; page 24, lines 12-16**).

Minor comments:

1. Please go through the manuscript and polish English and grammar. For example,

* Page 3: Line 3: "study"  "studies"

* Page 3: Lines 6-7: "a liquid chromatography coupled-mass spectrometry and long-read sequencing was"  "a liquid chromatography coupled-mass spectrometry and long-read sequencing were"

* Page 5: Line 9: delete "in"

* Page 5: Line 12-13: change "or" to "and" or "have" to "has"

* Page 5: Line 18: "complication"  "complications"

* Page 6: Line 4-5: "With proliferation of these pathogenic bacteria come production of uremic toxins"  "With the proliferation of these pathogenic bacteria comes the production of uremic toxins"

...

* Page 21: Line 3: "The quantity 353 of sequencing reads was evaluated t using Microbial Genomics Module"  delete "t"

* Page 21: Line 9: "algorism"  "algorithm"

* Page 23: Line 1: "sofeware"  "software"

Response:

The manuscript has been checked and edited by native English editor to eliminate typo and grammatical error prior to the revision.

November 9, 2022

Prof. Jung-Chun Lin
Taipei Medical University
School of Medical Laboratory Science and Biotechnology, College of Medical Science and Technology
250 Wu-Hsing Street
Taipei 110
Taiwan

Re: Spectrum02805-22R1 (Exploring the relevance between gut microenvironment and chronic kidney disease with distinct pathogenic factor)

Dear Prof. Jung-Chun Lin:

Thank you for submitting your manuscript to Microbiology Spectrum. Now your manuscript has been evaluated by two external experts, although they both find your study interesting, one reviewer insisted that you should make public both the raw data and codes for the data analyses. I tend to agree with him/her since we have an open data policy. Ideally, you can upload your data to a public database such as the NCBI or ENA, and your codes for analyzing the processed data to a Github depository or something similar.

Link Not Available

Sincerely,

Wei-Hua Chen

Journals Department
Reviewer comments:

Reviewer #1 (Comments for the Author):

The authors have addressed the issues in detail in the revised version of the manuscript appropriately. There are no further comments or suggestions for the authors.

Reviewer #2 (Comments for the Author):

- * The authors tried to reflect the major issues raised by the reviewer as much as possible.
- * However, the dataset is still restricted and could be available per request that is not a public dataset.

Staff Comments:

Preparing Revision Guidelines

Please return the manuscript within 60 days; if you cannot complete the modification within this time period, please contact me. If you do not wish to modify the manuscript and prefer to submit it to another journal, please notify me of your decision immediately so that the manuscript may be formally withdrawn from consideration by Microbiology Spectrum.

Response to Reviewers' Comments
(Manuscript Number: Spectrum02805-22R1)

Dear Editor:

We thank you for your response and for allowing revision of our manuscript (Spectrum02805-22R1; *Exploring the relevance between gut microenvironment and chronic kidney disease with distinct pathogenic factor*). The manuscript was revised in line with the valuable suggestions and comments of the reviewer. We hope that the revised manuscript achieves reviewer satisfaction. Our point-by-point responses to all specific reviewer comments, suggestions, and queries are as follows.

Response to Editor's comments

Thank you for submitting your manuscript to Microbiology Spectrum. Now your manuscript has been evaluated by two external experts, although they both find your study interesting, one reviewer insisted that you should make public both the raw data and codes for the data analyses. I tend to agree with him/her since we have an open data policy. Ideally, you can upload your data to a public database such as the NCBI or ENA, and your codes for analyzing the processed data to a Github depository or something similar.

Response:

Raw 16S rRNA sequencing data sets were deposited with the NCBI BioProject database under accession number PRJNA899930 according to the Editor's comment (**page 26, lines 1-2**). The analytic workflow toward the taxonomic identification and related statistical analyses using commercialized programs have been described within the **Materials and Methods** section. The metabolite analyses were commissioned to a commercial company and the publicity of analytic workflow or program is not available.

Response to Reviewer 2's comments

1. The authors tried to reflect the major issues raised by the reviewer as much as possible. However, the dataset is still restricted and could be available per request that is not a public dataset.

Response:

Raw 16S rRNA sequencing data sets were deposited with the NCBI BioProject database under accession number PRJNA899930 according to the reviewer's comment (**page 26, lines 1-2**).

November 22, 2022

Prof. Jung-Chun Lin
Taipei Medical University
School of Medical Laboratory Science and Biotechnology, College of Medical Science and Technology
250 Wu-Hsing Street
Taipei 110
Taiwan

Re: Spectrum02805-22R2 (Exploring the relevance between gut microenvironment and chronic kidney disease with distinct pathogenic factor)

Dear Prof. Jung-Chun Lin:

Congratulations. Your manuscript has been accepted, and I am forwarding it to the ASM Journals Department for publication. You will be notified when your proofs are ready to be viewed.

Sincerely,

Wei-Hua Chen
Editor, Microbiology Spectrum
